# REVISITING ON-POLICY
# DEEP REINFORCEMENT LEARNING

## ABSTRACT

On-policy Reinforcement Learning (RL) offers desirable features such as stable learning, fewer policy updates, and the ability to evaluate a policy's return during training. While recent efforts have focused on off-policy methods, achieving significant advancements, Proximal Policy Optimization (PPO) remains the go-to algorithm for on-policy RL due to its apparent simplicity and effectiveness. However, despite its apparent simplicity, PPO is highly sensitive to hyperparameters and depends on subtle and poorly documented tweaks that can make or break its success–hindering its applicability in complex problems. In this paper, we revisit on-policy deep RL with a focus on improving PPO, by introducing principled solutions that enhance its performance while eliminating the need for extensive hyperparameter tuning and implementation-level optimizations. Our effort leads to PPO+, a methodical adaptation of the PPO algorithm that adheres closer to its theoretical foundations. PPO+ sets a new state-of-the-art for on-policy RL on MuJoCo control problems while maintaining a straightforward trick-free implementation. Beyond just performance, our findings offer a fresh perspective on on-policy RL that could reignite interest in these approaches.

## 1 INTRODUCTION

A fundamental distinction in Reinforcement Learning (RL) lies between on-policy and off-policy methods (Sutton and Barto, 2018). On-policy methods, such as Proximal Policy Optimization (PPO) (Schulman et al., 2017b) and Trust-Region Policy Optimization (TRPO) (Schulman et al., 2015), directly optimize the expected reward under the current policy's state-action distribution, improving the policy while actively interacting with the environment. This leads to stable learning and safer exploration since the policy stays close to the data distribution it learns from, though at the cost of potentially reduced sample-efficiency. In contrast, off-policy methods optimize the expected reward under a different distribution—often using an exploration or behavior policy. By leveraging data generated from different policies, off-policy methods can reuse past experiences, boosting sample-efficiency. This flexibility supports more aggressive exploration, making off-policy methods more suitable when data collection is expensive or restricted.

Recent advancements in off-policy approaches, such as Soft Actor-Critic (SAC) (Haarnoja et al., 2018) and Twin Delayed Deep Deterministic Policy Gradients (TD3) (Fujimoto et al., 2018), have significantly improved continuous control on complex tasks. However, on-policy algorithms have not kept pace in terms of asymptotic performance and sample-efficiency. While PPO remains a dominant choice in on-policy RL, delivering impressive results across a range of applications (Berner et al., 2019; Andrychowicz et al., 2020b; Mirhoseini et al., 2021; Rudin et al., 2022), it is hindered by the complexity of its inherent mechanisms, including trust-region optimization, multiple loss functions, and various implementation-specific optimizations, making it highly sensitive to hyperparameter tuning (Andrychowicz et al., 2020a; Huang et al., 2022).

Moreover, common practices in the use of PPO have crucial shortcomings. For example, despite the empirically demonstrated success of maximum entropy RL (Haarnoja et al., 2018; Bhatt et al., 2024) and theoretical works suggesting it can enhance the convergence of policy gradient methods (Mei et al., 2020; Cen et al., 2024), its application for on-policy deep RL remains underexplored. Additionally, common on-policy algorithms that utilize the policy gradient theorem frequently over-

look the discount factor in the state distribution. This omission is technically incorrect and can result in degenerate learning behaviors in certain environments (Thomas, 2014; Nota and Thomas, 2019).

These challenges, combined with the inherent complexity of current on-policy deep RL methods, motivate us to pursue simpler and more sample-efficient alternatives. In this paper, we introduce PPO+, a principled enhancement of the PPO algorithm that introduce targeted solutions to tackle PPO's drawbacks while eliminating the need of extensive hyperparameter tuning and subtle implementation-level optimizations. More concretely, we propose and demonstrate that leveraging off-policy data can significantly improve critic learning while preserving the on-policy formulation of the policy gradient. Additionally, we integrate recent advances in critic learning, such as those proposed by Bhatt et al. (2024), to further enhance performance. Furthermore, we reformulate the PPO optimization problem under the maximum entropy RL perspective for enhanced exploration. Finally, we address a key limitation of biased policy gradient estimates caused by improper discounting, which can adversely impact the performance of policy-gradient methods.

We show that PPO+ achieves state-of-the-art performance among on-policy methods for continuous control while maintaining a simple and trick-free implementation and being closely aligned with the theoretical foundations of on-policy RL.

## 2 BACKGROUND

### 2.1 ON-POLICY REINFORCEMENT LEARNING

Reinforcement Learning (RL) (Sutton and Barto, 2018) deals with the problem of an agent interacting with an environment to learn a policy that maximizes its return. Mathematically, an RL problem can be formulated as a Markov Decision Process (MDP) (Puterman, 1990), which is a tuple $\langle S, A, P, R, \mu_0, \gamma \rangle$, where $S \in \mathbb{R}^m$ is a continuous set of states and $A \in \mathbb{R}^d$ is a continuous set of actions. $P : S \times A \to \Delta S$ is the transition probability function[1], where $P(s'|s, a)$ denotes the probability of transitioning to state $s'$ after taking action $a$ in state $s$. $R : S \times A \to \mathbb{R}$ is the reward function, where $r(s, a)$ is the immediate reward received by the agent for taking action $a$ in state $s$. $\mu_0 \in \Delta S$ is the initial state distribution. $\gamma \in [0, 1)$ is the discount factor, which determines the importance of future rewards compared to immediate rewards.

In on-policy RL, the agent's goal is to learn a stochastic policy $\pi : S \to \Delta A$, that maximizes its expected discounted return $J(\pi) = \frac{1}{1-\gamma}\mathbb{E}_{s \sim d_\gamma^\pi, a \sim \pi}[r(s, a)]$, where we denote $d_\gamma^\pi(s) \triangleq (1 - \gamma)\sum_{t=0}^{\infty} \gamma^t P(s_t = s)$ the discounted state visitation density of the state $s$ under the policy $\pi$. This is in contrast to off-policy RL where the objective of the agent is to maximize the policy return under a different behaviour policy $\beta(a|s) \neq \pi(a|s)$ making the objective to maximize $J^\beta(\pi) = \frac{1}{1-\gamma}\mathbb{E}_{s \sim \rho_\gamma^\beta, a \sim \pi}[r(s, a)]$.

### 2.2 MAXIMUM ENTROPY REINFORCEMENT LEARNING

Traditional RL algorithms focus solely on maximizing the expected reward. However, this can lead to overly deterministic policies that may not be robust to unforeseen changes in the environment. Maximum entropy RL (Ziebart, 2010; Haarnoja et al., 2018) address this issue by incorporating an entropy bonus into the objective function. The entropy of a policy $\pi$ is a measure of its diversity or randomness and is defined as $H(\pi(.|s)) = -\sum_a \pi(a|s) \log \pi(a|s)$. By adding the entropy to the original reward, the agent is incentivized explicitly to explore while not sacrificing on the policy return. This is achieved by introducing a temperature parameter $\alpha$ and reformulating the objective function as

$$J(\pi) = \mathbb{E}_\pi \left[ \sum_{t=0}^{\infty} \gamma^t \left( r(s_t, a_t) + \alpha H(\pi(.|s_t))) \right) \right]. \tag{1}$$

The temperature $\alpha$ controls the trade-off between maximizing reward and entropy. A larger $\alpha$ leads to a greater emphasis on exploration and mode diversity in the policy. In practice, we observe that it considerably improves exploration and hence learning speed over state-of-art methods that optimize the conventional RL objective function (Schulman et al., 2017a).

---

[1]$\Delta X$ denotes the set of probability measures over a set $X$.

## 2.3 TRUST REGION METHODS

Initially introduced by Schulman et al. (2015), trust region deep RL methods are on-policy algorithms that optimize a surrogate objective by maximizing a lower bound on the policy return. Trust Region Policy Optimization (TRPO) constrains the policy update by limiting the KL divergence between the new policy $\pi'$ and the old policy $\pi$, ensuring updates remain within a "trusted region" for stable learning. However, TRPO's approach originally relied on a heuristic to enforce this constraint.

In Achiam et al. (2017), the authors formalized this heuristic by bounding the difference between the returns of two policies, $\pi'$ and $\pi$, as follows

$$J(\pi') - J(\pi) \geq \frac{1}{1-\gamma}\mathbb{E}_{s\sim d^\pi, a\sim \pi'}\left[A^\pi(s,a)\right] - \frac{2\gamma\epsilon^{\pi'}}{1-\gamma}\sqrt{\frac{1}{2}\mathbb{E}_{s\sim d^\pi}\left[D_{KL}(\pi'\|\pi)[s]\right]}, \quad (2)$$

where $\epsilon^{\pi'} \doteq \max_s |\mathbb{E}_{a\sim\pi'}\left[A^\pi(s,a)\right]|$.

By squaring the penalty term and applying the importance sampling trick to replace the expectation over $a \sim \pi'$ with $a \sim \pi$, this optimization problem can be rewritten as

$$\text{maximize}_{\pi'}\mathbb{E}_{s\sim d^\pi, a\sim\pi}\left[\frac{\pi'(a|s)}{\pi(a|s)}A^\pi(s,a)\right] \quad (3)$$

$$\text{subject to}\quad \mathbb{E}_{s\sim d^\pi}\left[D_{KL}(\pi'\|\pi)[s]\right] \leq \delta. \quad (4)$$

TRPO solves this optimization problem by approximating the KL divergence constraint using a second-order method involving the Fisher information matrix, which requires a conjugate gradient method for optimization. While this guarantees updates stay within a trusted region, making the learning process stable, it also makes the algorithm computationally expensive due to the need for calculating the Fisher information matrix and solving the constrained optimization.

To address this complexity, Schulman et al. (2017b) propose Proximal Policy Optimization (PPO), which simplifies the enforcement of the trust region by introducing a clipping mechanism. Instead of explicitly constraining the KL divergence, PPO limits the probability ratio between the new and old policies, ensuring updates remain moderate. This approach is simpler to implement and significantly reduces computational overhead while retaining stable learning performance.

## 2.4 ACTOR-CRITIC METHODS

Actor-critic methods are a class of RL algorithms consisting of an actor and a critic. The critic estimates policy performance, represented by the long-term action-value function $Q^\pi(s,a) \triangleq \mathbb{E}_{s'\sim d_\gamma^\pi, a'\sim\pi(s')}\left[r(s,a) \mid s_0 = s, a_0 = a\right]$ or the value function $V^\pi(s) \triangleq \mathbb{E}_{a\sim\pi(s)}\left[Q^\pi(s,a) \mid s_0 = s\right]$. The actor updates its parameters to maximize the policy return according to the critic, enabling more efficient learning than methods relying on Monte-Carlo estimates.

Actor-critic algorithms improve a parametric model of the critic and policy Sutton et al. (1999), typically implemented using neural networks, via gradient ascent. Temporal Difference (TD) learning, as described by Sutton (1988); Sutton and Barto (2018), provides an iterative method to estimate the action-value function $Q^\pi$ for policy $\pi$. The TD error is defined as

$$\delta_t = r_{t+1} + \gamma\widehat{Q}^\pi(s_{t+1}, a_{t+1}) - \widehat{Q}^\pi(s_t, a_t),$$

where $r_{t+1}$ is the reward after transition, $\gamma$ is the discount factor, and $s_t, a_t$ and $s_{t+1}, a_{t+1}$ are the current and next state-action pairs, respectively. The TD error $\delta_t$ serves as a learning signal for updating the action-value function, a key component of many RL algorithms, including $Q$-learning where $a_{t+1} = \arg\max_a \widehat{Q}^*(s,a)$ and SARSA where $a_{t+1} \sim \pi(s)$. The action-value function $Q^\pi$ is updated to minimize the TD error, allowing updates based on the difference between the estimated values of the next and current state-action pairs.

Traditionally, the critic $\widehat{Q}^\pi$ can be on-policy if data comes exclusively from policy $\pi$, i.e., SARSA algorithm (Sutton and Barto, 2018). Alternatively, we can use previously collected data as in DDPG, TD3 or SAC Haarnoja et al. (2018); Fujimoto et al. (2018), in which case $\widehat{Q}^\pi$ is trained off-policy.

## 3 ON THE LIMITATIONS OF PROXIMAL POLICY OPTIMIZATION

While Proximal Policy Optimization (PPO) (Schulman et al., 2017b) is a popular choice in RL due to its simplicity and stability compared to earlier methods like TRPO (Schulman et al., 2015), it still suffers from significant limitations under certain conditions.

Figure 1 shows empirical evidence of these limitations. For starters, normalization of rewards or advantage functions plays a critical role in stabilizing PPO's learning process. Without it, PPO often fails to learn effective policies, especially in environments where rewards have different magnitudes like Hopper or Walker2d. Moreover, we show that PPO performs poorly when using full-batch updates instead of mini-batches which is counterintuitive for an on-policy method. For our experiments, we perform the same number of updates to the PPO objective while using the full batch instead of the minibatch updates. This should in theory improve the performance of PPO as the critic and the estimated surrogate objective should be a better estimate of their respective ground truth. However, surprisingly the learning of PPO seems to collapse when this is done. We believe this deterioration happens because of the additional exploration encouraged by the noisier gradients due to minibatching.

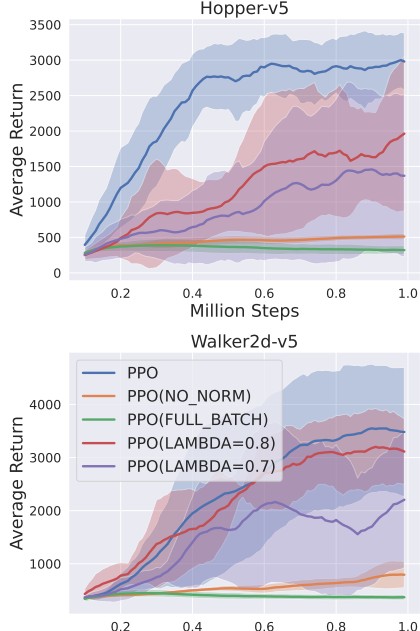

Finally, PPO exhibits performance degradation when the GAE-$\lambda$ is set to low values compared to its standard 0.95, essentially reducing the advantage estimate to a Monte-Carlo estimate. As we decrease the $\lambda$ value, the estimated advantages become much less accurate, hindering learning any useful policy. This sensitivity shows how PPO works only in regimes of high $\lambda$, which makes us question the quality of the value estimates obtained by the critic.

Figure 1: Sensitivity of PPO to reward normalization (NO_NORM), full-batch updates (FULL_BATCH) and the GAE-$\lambda$ (LAMBDA=$\{0.8, 0.7\}$).

Despite its widespread use (Berner et al., 2019; Andrychowicz et al., 2020b; Mirhoseini et al., 2021; Rudin et al., 2022), PPO's sensitivity to the GAE-$\lambda$, normalization, and minibatching, point to serious shortcomings of the algorithm. These limitations hint that further improvements are possible with the hope of improving the performance of deep on-policy methods.

## 4 ENHANCING PROXIMAL POLICY OPTIMIZATION: PPO+

The current landscape of deep on-policy RL methods highlights several fundamental issues, motivating a closer examination of how well existing approaches align with their theoretical foundations. In this section, we present and analyze *three* key methodological innovations for deep on-policy RL, which culminate in the development of our novel algorithm, PPO+ (Algorithm 1). Our aim with this new algorithm is to establish a more principled framework that rigorously adheres to the theoretical formulation of on-policy RL.

### 4.1 PROPERLY DISCOUNTING THE POLICY GRADIENT

The surrogate on-policy objective in (Equation (4)), describes the change in the discounted policy return in relation to the accumulated advantage over the discounted occupancy measure $d_\gamma^\pi$. Despite this, the majority of policy gradient methods bypass the use of the discounted state distribution when computing the policy gradient, opting to average the gradients across states instead. However, this practice results in a biased gradient estimator as it does not optimize the discounted objective.

Research has shown that this averaged gradient does not embody the gradient of any function (Nota and Thomas, 2019). As a result, there is no guarantee that algorithms following this direction will

---

**Algorithm 1** One Step of PPO+

---

**Require:** Current actor parameters $\phi$, critic parameters $\theta_1$, $\theta_2$, critic replay buffer $\mathcal{B}$

1: $\gamma_t = 1$               ▷ Initialize discount for proper discounting
2: $\mathcal{D} \leftarrow \emptyset$                  ▷ Reset the actor replay buffer
3: **for** $N_e$ episodes **do**
4:   $s_0 \sim \mu_0(s)$                ▷ Sample the initial state
5:   **for** each environment step **do**
6:    $a_t \sim \pi_\phi(a_t|s_t)$            ▷ Sample action from the policy
7:    $s_{t+1} \sim p(s_{t+1}|s_t, a_t)$      ▷ Sample transition from the environment
8:    $\mathcal{D} \leftarrow \mathcal{D} \cup \{s_t, a_t, \gamma_t\}$        ▷ Update the actor replay buffer
9:    $\mathcal{B} \leftarrow \mathcal{B} \cup \{s_t, a_t, r_t, s_{t+1}\}$      ▷ Update the critic replay buffer
10:    $\gamma_t \leftarrow \gamma_t \times \gamma, s_t = s_{t+1}$      ▷ Update state and discount factor
11:   **end for**
12:   $\gamma_t \leftarrow 1$
13: **end for**
14: **for** $N_u$ update steps **do**
15:   $B \leftarrow \{s, a, r, s'\} \sim \mathcal{U}(\mathcal{B})$      ▷ Sample a batch of off-policy transitions
16:   $y_i(s, a) = r + \gamma(Q_{\theta_i}(s', a') - \log \pi_\phi(a'|s'))$, $a' \sim \pi_\phi(.|s')$   ▷ Compute critic targets
17:   $\nabla_{\theta_i} \frac{1}{|B|} \sum_{(s,a,r,s') \in B} (Q_{\theta_i}(s, a) - y_i(s, a))^2$ for $i = 1, 2$   ▷ Update the critic networks
18: **end for**
19: $\hat{V}_i(s) = \mathbb{E}_{a \sim \pi} \left[ \hat{Q}_{\theta_i}^\pi(s, a) \right], \forall s \in \mathcal{D}$, for $i = 1, 2$    ▷ Compute value function estimates
20: $\hat{A}^\pi(s, a) = \frac{1}{2} \sum_{i \in 1,2} \hat{Q}_{\theta_i}^\pi(s, a) - \hat{V}_i(s), \forall s, a \in \mathcal{D}$   ▷ Compute advantage function estimates
21: $\phi = \arg\max_{\phi'} \sum_{(s_t, a_t, \gamma_t) \in \mathcal{D}} \gamma_t \min \left( \frac{\pi_{\phi'}(a_t|s_t)}{\pi_\phi(a_t|s_t)} \hat{A}^\pi(s_t, a_t), \text{clip} \left( \epsilon, \hat{A}^\pi(s_t, a_t) \right) \right)$
22:
23: **return** $\phi, \theta_1, \theta_2$              ▷ Optimized parameters

---

converge to a 'reasonable' optimum. In fact, it is possible to construct a counterexample where the fixed point is globally pessimal for both the discounted and undiscounted objectives (Nota and Thomas, 2019). Despite these shortcomings, this estimator remains the most widely used for estimating the policy gradient, primarily due to its proven effectiveness in practical applications (Schulman et al., 2017b; Haarnoja et al., 2018; Fujimoto et al., 2018). Hence, to adhere to the theory of RL, and to make sure to optimize for a valid objective, we use the discounted state distribution $d_\gamma^\pi$ for our policy gradient.

## 4.2 OFF-POLICY CRITIC LEARNING

Temporal-Difference (TD) learning, as outlined by Sutton and Barto (2018), offers a methodology for learning the value function using only system transitions, as expressed in Equation (2.4). This algorithm is a cornerstone in the field of RL, with extensive research dedicated to understanding its properties. It is well known that when TD is applied to a tabular value function representation, it converges to the true value function (Dayan, 1992; Jaakkola et al., 1993). Conversely, on-policy TD learning approaches using linear function approximation have been proven to converge to a fixed point in the vicinity of the projection of the true value function (Tsitsiklis and Van Roy, 1996).

However, divergence may occur with standard TD learning when states are sampled off-policy and linear function approximation is used (Baird, 1995). This issue has prompted the creation of several alternative algorithms specifically engineered to guarantee convergence under off-policy sampling (Kolter, 2011; Diddigi et al., 2019). In light of the lack of convergence guarantees for simple off-policy TD learning, the desirable properties of on-policy TD learning have inspired the development of deep RL on-policy methods that learn a critic $Q^\pi$ using exclusively the data generated by $\pi$, forgoing the use of a replay buffer to store previous transitions (Schulman et al., 2015; 2017b).

Despite the known limitations of TD with off-policy data, there has been notable success in using off-policy data to train critics in both online algorithms (Lillicrap, 2015; Haarnoja et al., 2018; Fujimoto et al., 2018) and most of the offline RL approaches (Wu et al., 2019; Kumar et al., 2019; Fujimoto and Gu, 2021). Surprisingly, this strategy has not yet been explored for on-policy al-

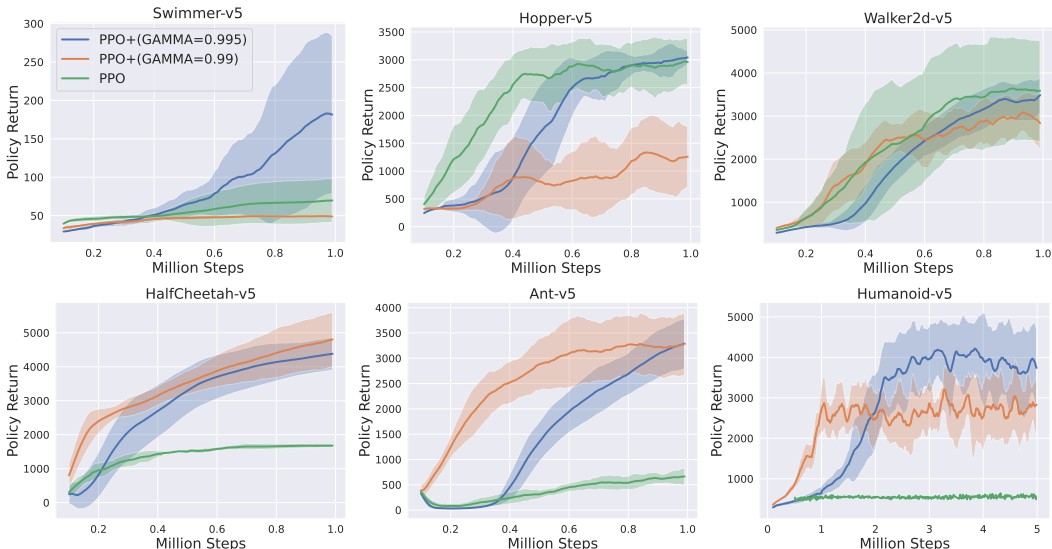

Figure 2: Evolution of the undiscounted policy return on the MuJoCo-v5 tasks. We use 10 random seeds for every algorithm and show the standard deviation.

gorithms. We hypothesize that leveraging off-policy data to improve critic approximation could enhance the accuracy of on-policy gradient estimates, potentially leading to better performance.

Indeed, when looking closer, one of the primary factors contributing to this non-convergence is state aliasing, a phenomenon that occurs in off-policy approximation when the function approximator perceives different states as identical, leading to information loss and potential divergence in learning (Sutton et al., 2016). Theoretically, the bias introduced by off-policy approximation diminishes with larger regressors (Sutton et al., 2016). This is attributed to the ability of larger regressors to capture more nuances in the state representation, thereby reducing the likelihood of state aliasing. However, it is crucial to note that while larger regressors can mitigate bias, they may concurrently increase the variance of the estimates. By using this insight, we choose to integrate off-policy data into our critic learning scheme by keeping track of past transitions via a replay buffer.

### 4.3 MAXIMUM ENTROPY FOR ON-POLICY REINFORCEMENT LEARNING

Maximum entropy RL augments the classic training objective with an additional term that encourages exploration and has proven successful in off-policy scenarios. However, surprisingly, it remains largely unexplored in on-policy RL. As no theoretical or technical limitations prevent us from using the maximum entropy formulation, we use it in PPO+ following the SAC update of the critic and the actor (Haarnoja et al., 2018).

## 5 EXPERIMENTAL VALIDATION

We empirically evaluate PPO+ against PPO on the MuJoCo benchmark for continuous control (Todorov et al., 2012). Inspired by Bhatt et al. (2024), we use an ensemble of two critics, with an update-to-data ratio of 1:1 and without target networks. Our critics are trained independently (i.e., the TD target is not the minimum of two as in Haarnoja et al. (2018); Fujimoto et al. (2018)) and only used to improve the quality of the estimate by averaging them. Since optimizing for the discounted objective increases the sensitivity of undiscounted performance to the choice of discount factor, we present results for two variants: PPO+ ($\gamma = 0.995$), where the policy is updated every 5000 steps, and PPO+ ($\gamma = 0.99$), which uses a more typical discount factor of $\gamma = 0.99$ and updates the actor every 2000 environment interactions. We report all the hyperparameters for our experiments are in Appendix A. For a detailed description of the differences in the implementation of PPO+ and PPO, we refer the reader to Table 2 in the Appendix B.

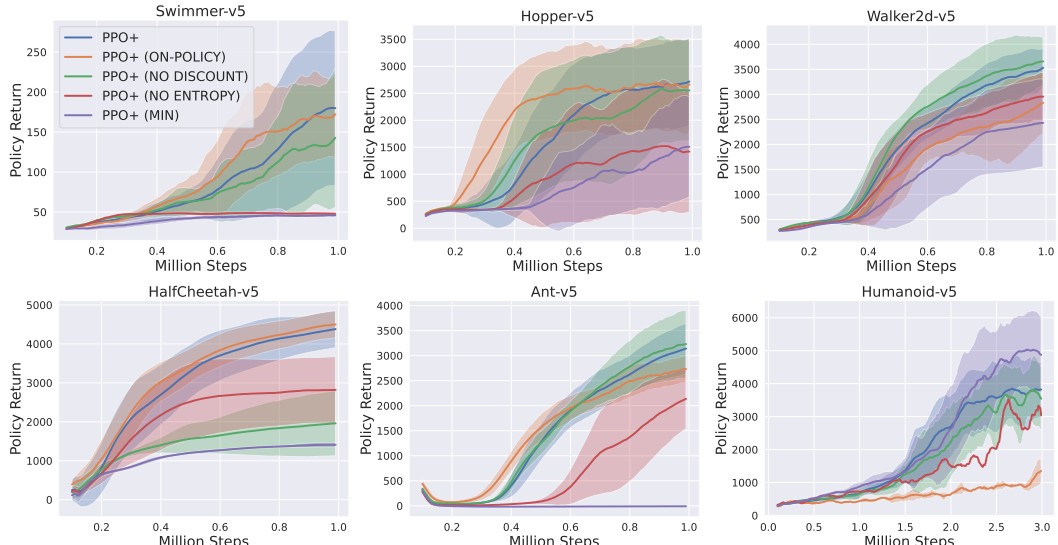

Figure 3: Evolution of the policy return on the MuJoCo-v5 tasks for various design choices. We use 10 random seeds for every algorithm. For all plots we use PPO+ ($\gamma = 0.995$). PPO+ (ON-POLICY) restricts the critics to using only on-policy data. PPO+ (NO DISCOUNT) foregoes discounting the surrogate objective. PPO+ (NO ENTROPY) removes the entropy bonus from the critics. PPO+ (MIN) uses the minimum of two critics as a target, as in TD3 (Fujimoto et al., 2018).

As shown in Figure 2, PPO+ ($\gamma = 0.995$) matches PPO's performance on two tasks and surpasses it on the remaining four, showing a notable performance gap in higher-dimensional tasks like Ant-v5 and Humanoid-v5. We believe the gap grows with the dimensionality of the problem because the advantage estimates of PPO are closer to those of REINFORCE with a baseline due to the high $\lambda = 0.95$ used. While PPO seems to work fine for low-dimensional tasks, estimating advantages from Monte-Carlo returns seems to work less as the dimensionality of the task and the bootstrapping inherent to TD-learning seem to outperform it clearly.

Indeed, TD-learning is more sample-efficient than REINFORCE in high-dimensional problems due to its use of bootstrapping, allowing for updates from partial rollouts rather than full trajectories. This helps reduce variance in gradient estimates, which is crucial for limited samples. Additionally, TD-learning supports "trajectory stitching", where updates integrate information from different parts of trajectories, combining insights from multiple paths. REINFORCE, by contrast, relies on full trajectories, making it less efficient and higher variance, especially in the case of high dimensional problems. Furthermore, as demonstrated earlier in Figure 1, simply removing one code-level optimization (e.g., reward normalization) allows PPO+ to significantly outperform PPO across all tasks. Importantly, PPO+ achieves this without relying on any code-level optimizations.

## 5.1 ON THE IMPACT OF PPO+ ENHANCEMENTS

In Figure 3, we present an ablation study for PPO+ examining our three key design choices: **(1)** the application of the true discounted policy gradient; **(2)** the use of off-policy data for training the critic; and **(3)** the use of the maximum entropy objective. Our results demonstrate that restricting the critic's training to on-policy data significantly degrades performance, even impeding learning in tasks such as Humanoid-v5. Overall, we find that training the critic on larger datasets, even with off-policy data, is generally advantageous compared to limiting the training to a smaller pool of freshly generated data. This is in contrast to the common practice of restricting training the critic to on-policy data for on-policy gradient methods. Interestingly, while the use of off-policy data is already well explored in deep off-policy actor-critic methods like Lillicrap (2015); Haarnoja et al. (2018); Fujimoto et al. (2018), it is not clear whether this choice majorly benefit the actor or the critic. This work suggests that at least the critic has a great benefit from the use of off-policy data, showing that deep neural networks can overcome the state aliasing inherent to off-policy TD learning.

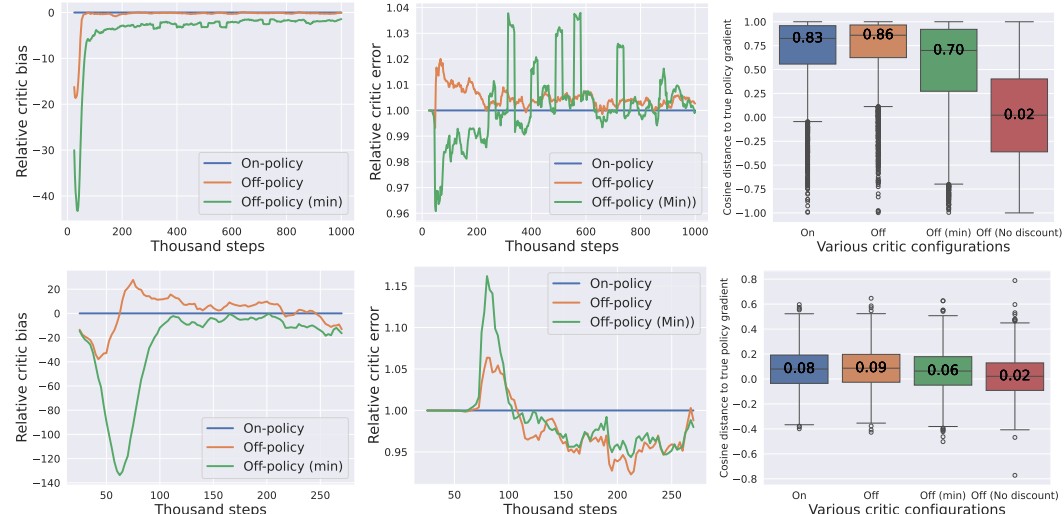

Figure 4: **Left:** Evolution of the bias difference across various LQR tasks, using the bias from the "On-policy" configuration as the reference. **Middle:** Evolution of the ratio of test error relative to the "On-policy" configuration. **Right:** Distribution of the cosine similarity between the estimated gradients and the true policy gradient across different configurations. **Top:** 2-dimensional LQR; **Bottom:** 33-dimensional LQR.

Regarding the discounted policy gradient, we observe that applying a discount has little to no effect on certain tasks like Walker2d-v5 and Ant-v5, leads to performance improvements on some others like Swimmer-v5, Humanoid-v5, and has a significant impact in HalfCheetah-v5. We believe this is because HalfCheetah-v5 resembles the synthetic task formulated in (Nota and Thomas, 2019), where the averaged policy gradient results in a degenerate local optimum. Our results are in contrast to the ones of Che et al. (2023b) concerning the $\gamma^t$ method, demonstrating that discounting the policy gradient result in consistently improved performance. We believe this finding is due to the improvements in the estimation of the surrogate objective via improvements the critic (and hence implicitly the policy gradient).

As for using the maximum entropy objective, we find that it improves our performance consistently across all tasks. We posit that PPO achieve good performance without entropy bonus as the poor quality of its critic, and hence the surrogate objective, result in unintentional additional exploration. Using the minimum of two critics seems to be detrimental for performance, we believe this is because it inhibits exploration in most tasks, with the exception of Humanoid-v5 which seems to benefit from the extra conservatism.

In summary, our results suggest that **(1)** the discounting of the gradient contributes to better learning; **(2)** off-policy TD learning in on-policy RL consistently enhances performance; **(3)** the entropy bonus provides clear benefit to PPO+ as opposed to PPO which is indifferent to the entropy bonus (albeit slightly different one), as reported in Andrychowicz et al. (2020a).

## 5.2 ON THE BENEFIT OF PPO+ ENHANCEMENTS

To further justify the results in Figure 3, we consider LQR environments. For each seed, we train an agent and plot every 2000 interactions using a separately generated on-policy dataset. We consider different critic configurations, namely on-policy critics, off-policy critics, two critics trained independently, and training critics with the minimum target, as introduced by (Fujimoto et al., 2018) and later adopted by SAC (Haarnoja et al., 2018) and follow-up works (Bhatt et al., 2024).

The left and middle plots illustrate the effects of various critic configurations on the approximation error using separate on-policy data. We observe that training with off-policy data does not introduce additional bias compared to using only on-policy data. However, the middle plot reveals that off-policy data reduces approximation error in high-dimensional tasks. The use of the minimum pre-

diction of two critics as a target does not significantly affect the approximation error but it degrades the quality of the policy gradient, as shown in the right plot. Moreover, the right plot demonstrates that not discounting the gradient deteriorates the correlation between the estimated gradient and the true discounted policy gradient. The cosine similarity between the gradients of two independently trained off-policy critics drops from 0.86 and 0.09, respectively, to a near-orthogonal value of 0.02.

In conclusion, these findings reinforce the observations in Figure 3 that discounting the gradient plays a crucial role, using off-policy data improves policy gradient estimation, and using two separate critics enhances the policy gradient's quality compared to using the minimum of two critics.

## 6 RELATED WORKS

Numerous studies underscore the sensitivity of current deep on-policy methods to hyperparameters and implementation details (Huang et al., 2022; Andrychowicz et al., 2020a), urging the community to simplify and close the gap between theory and practical implementation.

Ilyas et al. (2018) observe that the behavior of deep PG algorithms differs greatly from its motivating frameworks. Specifically, learned value estimators frequently fail to fit the true value function, and there is a poor correlation between gradient estimates and the 'true' gradient. In Nota and Thomas (2019), the authors demonstrate that the undiscounted policy gradient does not correspond to the gradient of any objective function. They also identify instances where this empirical gradient can be suboptimal for both discounted and undiscounted policy return. (Thomas, 2014) introduce the $\gamma^t$ method used to discount the gradient in our work, (Che et al., 2023b) refine this method by creating an estimator with lower variance. Aside a few exceptions (Tosatto et al., 2020; 2022a;b; Che et al., 2023a), proper discounting remains uncommon in the deep RL literature. Several works have explored training critics using off-policy data (Degris et al., 2012; Haarnoja et al., 2018; Fujimoto et al., 2018), with Bhatt et al. (2024) being one of the first to streamline the critic learning process by eliminating the need for target networks, which were initially popularized by Mnih (2013).

Entropy regularization plays a pivotal role in numerous deep RL algorithms (Haarnoja et al., 2018; Bhatt et al., 2024). In fact, the entropy of the policy acts as a regularizer shaping the objective landscape (Ahmed et al., 2019). The prevalent strategy regularizes the policy evaluation phase by supplementing the standard RL task objective with an entropy term. This method guides policies towards regions of higher expected trajectory entropy, a scheme often referred to as maximum entropy RL (Ziebart, 2010; Haarnoja et al., 2018), which is recognized for enhancing the exploration capabilities and robustness of policies by fostering stochasticity. Recent studies on policy gradient methods have highlighted the efficacy of maximum entropy RL in speeding up convergence (Mei et al., 2020; Ahmed et al., 2019; Cen et al., 2024).

## 7 DISCUSSION AND CONCLUSION

In this work, we introduced PPO+, a methodical enhancement of the Proximal Policy Optimization (PPO) algorithm. PPO+ rigorously adheres to the theoretical foundations of on-policy Reinforcement Learning (RL) while maintaining a simple, trick-free implementation. PPO+ introduces three key improvements over PPO, namely training the critic using off-policy data while maintaining the on-policy policy gradient formulation, using the true discounted policy gradient, and employing maximum entropy exploration. Moreover, by focusing on the quality of the critic approximation, and consequently the surrogate objective estimator, PPO+ avoids complex critic learning schemes and implementation-level optimizations. In practice, PPO+ eliminates the use intricate critic learning schemes used in common practices and obtains state-of-the-art performance for deep on-policy RL methods in MuJoCo locomotion problems. Thank to its simplicity and rigorous formulation, we believe that PPO+ offers an accessible and solid ground for future research on on-policy deep RL. **Limitations.** Despite its strengths, PPO+ does not match the performance of its off-policy counterparts, e.g., SAC (Haarnoja et al., 2018) or TD3 (Fujimoto et al., 2018). Nevertheless, we hope that the simplicity of PPO+ and the insights provided in our work will inspire further interest in on-policy methods. Potential directions for improvement include better strategies for correcting the off-policy distribution to improve critic learning, which could potentially be integrated into actor updates. Other directions may focus on improving critic learning itself, such as exploring validation criteria (Kallel et al., 2024) or improving the neuroplasticity of the critic (Nikishin et al., 2022).

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

## A    HYPERPARAMETERS

| Parameter | PPO+($\gamma = 0.995$) | PPO+($\gamma = 0.99$) |
|---|---|---|
| optimizer | Adam | Adam |
| learning rate | $3 \cdot 10^{-4}$ | $3 \cdot 10^{-4}$ |
| discount ($\gamma$) | 0.995 | 0.99 |
| replay buffer size | $5 \cdot 10^4$ | $2 \cdot 10^4$ |
| number of critics | 2 | 2 |
| LayerNorm | True | True |
| number of hidden layers (all networks) | 2 | 2 |
| number of hidden units per layer | 256 | 256 |
| number of samples per minibatch | 256 | 256 |
| temperature | 0.05 | 0.02 |
| nonlinearity | TanH | TanH |
| actor update interval | 5000 steps | 2000 steps |

Table 1: Hyperparameters for PPO+.

## B    DIFFERENCES BETWEEN PPO AND PPO+ IMPLEMENTATIONS

| Attribute | PPO | PPO+ |
|---|---|---|
| GAE-$\lambda$ critic | ✓ | - |
| Reward normalization | ✓ | - |
| Advantage normalization | ✓ | - |
| Learning rate scheduler | ✓ | - |
| Separate backbone* | - | ✓ |
| Discounted policy gradient | - | ✓ |
| Full batch actor updates | - | ✓ |
| Uses off-policy data | - | ✓ |
| Maximum entropy objective | - | ✓ |

Table 2: *: In the original PPO implementation, both the actor and critic share a common backbone. However, this design necessitates careful hand-tuning of the losses propagated to the shared backbone from the actor and critic heads. In contrast, PPO+employs a separate critic network, which not only eliminates the need for manual loss balancing but also enables significantly more frequent critic updates (on the order of thousands) compared to PPO's standard 10 updates. This increased update frequency improves the critic's performance by allowing for better convergence, while avoiding the risk of overfitting the surrogate objective often encountered in PPO with frequent policy updates.

