# OpenReview forum: "Revisiting On-Policy Deep Reinforcement Learning"
_ICLR.cc/2025/Conference — ICLR 2025 Conference Withdrawn Submission_

### Official Review · Reviewer_N7R9 · 2024-10-21

**Soundness:** 2
**Presentation:** 4
**Contribution:** 2
**Rating:** 3
**Confidence:** 4

**Summary:**

The paper aims to address the limitations of PPO by introducing a variant called PPO+. The authors propose several theoretically motivated modifications to reduce hyperparameter sensitivity and eliminate reliance on implementation-level tricks. PPO+ is designed to maintain the simplicity of PPO while aligning more closely with the theoretical principles of on-policy reinforcement learning. The paper evaluates PPO+ on MuJoCo benchmarks and claims state-of-the-art performance among on-policy methods.

**Strengths:**

1. **Clarity and Methodology**: The paper is clearly written and thoroughly explains the methodology behind PPO+. It provides pseudocode and hyperparameter details, making the algorithm implementation straightforward and moderately reproducible.
2. **Combination of Techniques**: Although the modifications themselves may not be entirely novel, the combination presented in PPO+ and its alignment with theoretical foundations is practical and beneficial for the RL community especially since its relatively easy to implement.
3. **Ablation Studies**: The ablations are well-constructed, demonstrating the effectiveness of the individual components introduced in PPO+.

**Weaknesses:**

1. **Limited Comparisons**: The paper's claims of state-of-the-art performance are only made relative to PPO. Including comparisons with other on-policy methods like VMPO (https://arxiv.org/abs/1909.12238) or SPO (https://arxiv.org/abs/2402.07963) would provide a more comprehensive context and strengthen the results as I do not believe you can make this claim given only PPO as a comparison.
2. **Backbone Usage**: PPO+ uses a separate backbone for the critic and actor networks, while PPO does not. Previous work by Andrychowicz et al. (2021) suggests that separate networks generally perform better. The lack of an ablation study comparing the shared and separate backbones in PPO+ raises serious concerns about the true source of performance gains.
3. **Hyperparameter Sensitivity Claims**: The authors don’t explicitly claim that PPO+ is less hyperparameter-sensitive than PPO however they state it as a motivating factor for the creation of it and the paper does not provide any empirical evidence to support this. Without testing the robustness across different environments or hyperparameter settings, it seems that PPO+ doesn’t necessarily address a core motivation.
4. **Limited Evaluation**: The evaluation is restricted to Mujoco environments. While this is a common benchmark, it is not sufficient to demonstrate that PPO+ consistently outperforms PPO or is more generalised. Testing in a broader set of environments, like grid-based or discrete action spaces, would provide more robust support for the authors' claims.
5. **Overfitting to MuJoCo**: Given the limited environment diversity and potential over-tuning for Mujoco tasks, it is unclear if PPO+ is truly an improvement over PPO or merely a set of optimisations tailored to a specific domain.

**Questions:**

### Questions

1. Could the authors provide more empirical evidence supporting that PPO+ is less hyperparameter-sensitive? Specifically, how does PPO+ perform across a range of hyperparameter settings compared to PPO across that same range? Additionally, does the introduction of new hyperparameters such as entropy regularisation, number of critics, replay buffer size now make it even harder to tune for new environments?
What is the effect of the CrossQ modifications? Do you use them with the PPO baseline as well. Is there an ablation of PPO+ without using the crossQ modifications?
3. Why was only MuJoCo used for evaluation? Would the authors be willing to extend their tests to additional benchmarks such as discrete action environments or grid-based tasks to validate the generalizability of PPO+?
4. Could an ablation study comparing the separate backbone used in PPO+ with a shared backbone approach be added to verify that the performance gains are due to the proposed modifications and not just architectural differences?

### Suggestions
1. **Use of Evaluation Methodology**: Consider using evaluation methodology like [rliable](https://github.com/google-research/rliable) to present more statistically robust results.
2. **Additional Comparisons**: Including at least one other on-policy algorithm (e.g., VMPO or SPO) would provide valuable context and strengthen the impact of the results.
3. **Diversify Environment Tests**: Extending the evaluation to other types of environments and presenting results where the hyperparameters are consistent across these tests could better support the claims of reduced hyperparameter sensitivity.

Ultimately, I liked the paper but I think without an ablation on the shared torso i.e. using one for PPO baseline, and without a different environment suite of results, i am not willing to accept the paper. Additionally, the use of crossQ modifications concerns me as we dont fully know the interaction of these modifications. Its possible a lot of the results come from here as well. If my core concerns are addressed, I’m willing to raise my score.

---

### Official Review · Reviewer_42xL · 2024-11-03

**Soundness:** 3
**Presentation:** 3
**Contribution:** 3
**Rating:** 6
**Confidence:** 2

**Summary:**

This paper introduces PPO+, a new on-policy deep reinforcement learning algorithm that builds upon and improves the Proximal Policy Optimization (PPO) algorithm. The authors identify several key limitations of PPO, including sensitivity to hyperparameters and deviations from the theoretical foundations of on-policy RL. They propose solutions to address these shortcomings, resulting in an algorithm that is more principled, robust, and achieves state-of-the-art performance for on-policy methods on MuJoCo control tasks. PPO+ incorporates three major improvements: (1) correct discounting in policy gradient computation, (2) integration of off-policy data for critic learning, and (3) maximum entropy regularization.

**Strengths:**

- The paper clearly identifies the limitations of PPO and motivates the need for a more principled approach.
- The paper thoroughly reviews relevant literature on on-policy RL, maximum entropy RL, trust region methods, and actor-critic methods, effectively placing the proposed approach within the existing litterature.
- PPO+ presents theoretically grounded modifications, such as leveraging off-policy data in on-policy settings, which could broaden the applicability of PPO and reduce the need for extensive tuning.
- Experimental results, especially on MuJoCo environments, show consistent improvements over PPO, suggesting that PPO+ delivers better results in continuous control tasks.
- The ablation studies strengthen the authors' claims, providing insights into how each enhancement (e.g., entropy regularization) impacts performance.

**Weaknesses:**

- The experiments are currently limited to MuJoCo control tasks. Evaluating PPO+ on a wider range of environments would provide more comprehensive proof of its capabilities.
- A discussion of the performance gap between PPO+ and off-policy counterparts would strengthen the paper.
- While the focus on PPO is understandable given its popularity, the paper would benefit from comparing PPO+ to other on-policy algorithms beyond PPO.
- The authors acknowledge that optimizing for the discounted objective increases sensitivity to the choice of discount factor. While they present results for two different discount factors, further investigation into this sensitivity and strategies for mitigating it would enhance the practicality of PPO+.

**Questions:**

- Could you elaborate on the performance gap between PPO+ and off-policy methods like SAC and TD3? What are the potential challenges and opportunities for bridging this gap within the on-policy framework?
- Have you considered evaluating PPO+ on other benchmark environments beyond MuJoCo control tasks?
- Given PPO+’s slight increase in complexity, do you have insights into how it compares in terms of training time relative to PPO, especially as task dimensionality increases?

---

### Official Review · Reviewer_v8vk · 2024-11-04

**Soundness:** 2
**Presentation:** 3
**Contribution:** 1
**Rating:** 3
**Confidence:** 4

**Summary:**

This paper introduces PPO+. PPO+ aims to augment the current PPO algorithm with best-known practical practices from well-known algorithms such as SAC and TD3, as well as theoretical principles, to improve the performance and sample efficiency of PPO. These features are: 1) using off-policy data by introducing a replay buffer, 2) learning a Q-function instead of only a value function, 3) using an entropy bonus, and 4) discounting the state distribution.

**Strengths:**

Paper is well written and easy to follow.

**Weaknesses:**

--This paper presents some interesting ideas, but I think it could be strengthened by highlighting its contributions more clearly. For example, incorporating off-policy data with a replay buffer and learning a Q-function instead of a state-value function shifts the algorithm towards the off-policy RL domain. To really showcase the algorithm's effectiveness, it would be beneficial to see comparisons with established off-policy algorithms like SAC and TD3. This would provide a clearer picture of its performance within the broader context of off-policy RL.

--It's also worth noting that one advantage of on-policy algorithms is their ability to learn by fitting only a value function, which can be simpler than fitting a Q-function. Introducing Q-learning in this context might add complexity, which seems to contrast with the authors' claim of increased simplicity. It would be helpful to see further discussion on this design choice and its potential implications in the context of on-policy RL.

--Adding an entropy bonus is a well-established technique, having been introduced in the original PPO paper. The entropy weight is already a standard hyperparameter in most PPO implementations. More discussion on how the use of the entropy bonus here differs from standard PPO would be helpful.

--Authors noted, reintroducing discounting to the state distribution doesn't yield significant performance improvements. A discussion on in which scenarios using a discounted state distribution would be beneficial would also be helpful.

--Finally, the experimental results presented aren't entirely conclusive. In some domains, PPO performs better than PPO+. It's more fair to compare PPO+ with off-policy algorithms. However, as the authors mentioned, their method doesn't currently outperform SAC or TD3, despite incorporating many of the components from those algorithms. This raises questions about the specific benefits and potential advantages of the proposed modifications.

**Questions:**

See above.

---

### Official Review · Reviewer_pDTs · 2024-11-05

**Soundness:** 3
**Presentation:** 3
**Contribution:** 2
**Rating:** 5
**Confidence:** 5

**Summary:**

The paper revisits on-policy RL, which is still one of the most predominant paradigms for learning controllers in simulation (or nowadays for RLHF of large models) since on-policy RL can give high quality (and minimally biased) policy improvement. The authors note that despite the simplicity of the theory underlying basic on-policy algorithms, in practice (partially due to the fact that on-policy algorithms have to trade-off optimization and exploration) they can be brittle/sensitive to hyperparameter settings.

The authors revisit the and robustify a popular on policy method (PPO) utilizing some of the insights from the recent literature on policy optimization; e.g. taking inspiration from recent results from the off-policy literature (i.e. SAC and others) such considering a maximum entropy formulation and learning an action-value (Q-function) critic instead of a state value function.

**Strengths:**

- The adjustments made to PPO are reasonably well motivated and pull directly from the existing literature on off-policy methods.
- There is a lack in the literature for good empirical evaluations of existing RL algorithms in fair comparisons; bridging the gap between on and off-policy methods (as done here) certainly fills part of this void.
- The stability of PPO is of high practical relevance for varying applications from control to RLHF of large models and thus any improvements are relevant to the community.

**Weaknesses:**

1. The experiments are unfortunately fairly limited in scope. Only 6 mujoco control domains are used and only two of them (and and humanoid) would be considered high dimensional in 2024. This limits the evidence that the paper can present for its suggested modifications seveerly.
2. The presentation of the experiments is  lacking:
2a: A comparison to baseline PPO is presented on two domains in Figure 1 and 2. With PPO failing on the high dimensional domains. This doesn't inspire huge confidence in the results. What is causing this? Is the asymptotic performance fine and the main difference is just the speed-up from the Q-function and standard PPO would just need to run much longer?
2b: Further Figure 3 ablates some choices of the algorithm but again seems lacking. We get no insight into which of the proposed modifications exactly makes things work. For example: how would standard PPO but with a Q-function do? It also seems like PPO without discounting could be fine on-policy (but we are missing those results here, i.e. the combination of on-policy and no discounting).
2c: A the practical implementations of PPO for any domain with higher dimensional observations (or larger models) might consider computing the loss only on a trajectory snippet extracted from a full episode. It is unclear how that would affect e.g. the discounting.
3. Out of the three proposed modifications two are already routinely considered in the literature/implementations: entropy regularization is a standard feature in many PPO implementations; using discounting for the 'policy gradient' loss has been considered multiple times in the literature (also partially noted by the authors) and not been consistently proven to make a big difference, so most implementations omit it. This leaves the reviewer thinking that the main contribution is to consider learning an action-value critic off-policy, but unfortunately the experiments do not properly ablate and compare this modification (see above).
4. In many applications it is generally hard to learn an action-value critic (since conditioning on high-dimensional actions comes with it's own problems) especially when dealing with large models and or large action spaces so the algorithm here may not be generally an improvement in all cases (e.g. the situation might look very different for RLHF of large models or for experiments requiring vision inputs).

**Questions:**

I do not have any direct questions to the authors aside from those listed in the weaknesses section above.
My main concern with the paper is the rigor of the experimental evaluation which in addition with a lack of novelty for the suggested improvements leave me wanting for clear conclusions I would trust after reading the paper.

---

### Official Review · Reviewer_VzKe · 2024-11-06

**Soundness:** 2
**Presentation:** 2
**Contribution:** 2
**Rating:** 3
**Confidence:** 3

**Summary:**

The paper proposes several modifications to the PPO algorithm: entropy regularization, off-policy value function learning, and discounting of the state distribution. It shows experimental results that investigate the effect of these modifications and compares them to a vanilla PPO implementation.

**Strengths:**

The paper is mostly clearly written. It proposes several reasonable, albeit well known, algorithmic components and integrates them into the PPO algorithm. It shows experimental results that suggest that these modifications can lead to improvements compared to a vanilla PPO baseline.

**Weaknesses:**

None of the proposed modifications is novel, they have all been well studied in the literature. The paper dedicates a significant amount of space to reviewing these fairly well known ideas. I don't think that merely putting them together in a new combination is in itself a significant contribution.

The experiments are not conclusive since important comparisons to SOTA off-policy algorithms are missing. Since the paper introduces effectively an off-policy component into the algorithm (with the need to implement a replay buffer etc.), I would have really liked to see this comparison. Indeed the authors state (in the limitations) that the proposed combination of algorithmic components underperforms such existing algorithms which begs the question why one should use the combination proposed in the present paper. (NB, some off policy algorithms such as MPO also use a trust region and in that respect bear similarities to PPO.)

For this to be a strong paper I would have expected an insightful discussion why the specific algorithmic combination should be particularly useful / interesting, a demonstration that it clearly outperforms existing algorithms on relevant problems, and a detailed analysis why this is the case.

**Questions:**

Some minor comments:

There seem to be several details missing, e.g. what implementation is used to produce the baseline for PPO; what is the actually benchmark that's being used (the paper generically cites Mujoco), etc.. (Apologies if I've missed these.)

Some citations are messed up (e.g. bottom of page 3).

The paragraph starting in line 294 on page 6 is not clear.

---

### Note · Authors · 2024-11-25

**Comment:**

Dear Reviewers,

In light of the feedback provided, we have decided to withdraw our paper.

We sincerely appreciate the thoughtful and constructive comments shared by the reviewers. These will be useful as we revise and strengthen the paper for future submissions.

Thank you for your time and consideration.

Best regards,

The Authors

**Withdrawal Confirmation:**

I have read and agree with the venue's withdrawal policy on behalf of myself and my co-authors.